# The Inhibition of Amylase and ACE Enzyme and the Reduction of Immunoreactivity of Sourdough Bread

**DOI:** 10.3390/foods9050656

**Published:** 2020-05-19

**Authors:** Anna Diowksz, Alicja Malik, Agnieszka Jaśniewska, Joanna Leszczyńska

**Affiliations:** 1Institute of Fermentation Technology and Microbiology, Faculty of Biotechnology and Food Sciences, Lodz University of Technology, 90-924 Lodz, Poland; anna.diowksz@p.lodz.pl (A.D.); agnieszka.jasniewska@p.lodz.pl (A.J.); 2Institute of Natural Products and Cosmetics, Faculty of Biotechnology and Food Sciences, Lodz University of Technology, 90-924 Lodz, Poland; alicja.malik@dokt.p.lodz.pl

**Keywords:** wheat sourdough bread, digestion, gluten, angiotensin-converting enzyme, alpha-amylase, immunoreactivity

## Abstract

This study examines the potential health benefits of different types of wheat sourdough bread against diseases of civilization. Celiac disease, diabetes and hypertension affect large numbers of the world’s population, increasing demand for novel treatments and ways of improving patient welfare. Different types of artisan breads were subjected to in vitro simulated digestion prior to analysis. The G12 test and ELISA with human sera were used for immunoreactivity analysis. The activity of α-amylase inhibitors and angiotensin-converting enzymes (ACE) was also assessed. The addition of sourdough to the analyzed wheat bread raised the content of α-amylase inhibitors and angiotensin-converting enzyme inhibitors while reducing their immunoreactivity. However, despite decreases in the antigenicity of the wheat flour proteins, the sera showed various reactions, depending on the individual patient’s susceptibility to gluten.

## 1. Introduction

Poor dietary habits and lack of physical activity are the main causes of diet-related conditions such as obesity, diabetes, cardiovascular diseases, some types of cancer, dental problems, osteoporosis and bone fractures, [1]. To lower the risk of such conditions, scientists recommend reducing the consumption of products which are high in saturated and trans fats, refined grains, sugar and salt, while increasing the intake of fresh fruit, vegetables and whole grains [2]. Consumption of fermented foods is also recommended. These can increase levels of antibodies and strengthen the immune system. Fermented foods can also regulate the appetite and reduce cravings for sugar and refined carbohydrates. Another benefit is that lacto-fermentation boosts the nutrient content of food and makes the minerals in fermented foods more readily accessible. Bacteria in cultured foods produce vitamins and enzymes that are valuable for digestion. Lactic acid bacteria (LAB) may even be able to lessen dietary problems in individuals living with diabetes, celiac disease or gluten intolerance, as well as have the potential to lower hypertension.

One disorder resulting from a poor diet is diabetes. This condition affects nearly 10% of the world’s population [3]. Typical treatment methods include insulin supplements and oral hypoglycemic agents, as well as appropriate diet and exercise. The α-glucosidase enzymes, such as α-amylase, break down starch and/or disaccharides into monosaccharides. Their inhibitors prolong and also delay carbohydrate digestion. Longer digestion times lead to a noticeable decrease in the rate of glucose absorption. This blunts the postprandial increase of plasma glucose [4].

Besides diabetes, poor dietary habits can result in hypertension. It is indirectly caused by the angiotensin-converting enzyme (ACE), which catalyzes the inactivation of bradykinins by acting on the kinin–kallikrein system. Bradykinins are compounds with extremely high vasodilatory action [5]. Angiotensin-converting enzymes also catalyze the production of Angiotensin II, which activates type 1 angiotensin receptors by binding to them. Such action stimulates the release of aldosterone, which promotes water and sodium reabsorption, leading to increased artery pressure [6]. Conversely, ACE-inhibitor (IACE) compounds will result in antihypertensive action [5].

Celiac disease is another diet-related condition manifested in genetically susceptible individuals. It is an immune-mediated enteropathy and affects the small intestine of the gastrointestinal tract and is characterized by an abnormal immunologic response triggered by the ingestion of gluten [7]. Gluten is composed of two protein fractions: alcohol-soluble gliadins and polymeric, alcohol-insoluble glutenins, which are rich in prolamine. Gliadins contribute to the cohesiveness and elasticity of dough, while glutenins contribute to its extension resistance [8]. Gluten peptides, such as 33-mer, are able to stimulate T cells, resulting in immune responses. At the same time, they are resistant to degradation by all gastric, pancreatic and intestinal brush border membrane proteases in the human intestine [9]. Typical symptoms of celiac disease include steatorrhea (diarrhea due to malabsorption of fat), weight loss, weakness and anemia [10]. The only treatment method for celiac disease is an elimination diet excluding gluten. There is, therefore, a great demand for a wider range of products suitable for celiac patients, including from bakeries and cereal-based food producers, who are interested in new approaches to reducing gluten levels so that their products are more celiac friendly.

Sourdough is a mixture of flour and water fermented with lactic acid bacteria and yeast. It may occur naturally or be pitched into the dough composition. Its metabolic activity has a beneficial effect on the character of the final product [11]. Processes such as lactic acid fermentation, the production of exopolysaccharides, proteolysis, and synthesis of volatile and microbial compounds can enhance the quality of bread by improving its volume, texture, flavor and nutritional value. They may also prolong its shelf life by slowing the staling process [12]. Sourdough technology is now being investigated as a new method for gluten degradation, as a result of the increased acidity caused by lactic acid bacteria. Acidification assists in the reduction of disulfide bonds in gluten [13].

The purpose of the present study was to examine the potential health properties of wheat sourdough bread produced by an artisan bakery in comparison to yeast bread.

## 2. Materials and Methods

Different types of wheat sourdough bread were analyzed in terms of the activity of amylase inhibitors and ACE activity, as well as for their immunoreactivity using a G12 test (an immunosorbent assay for detection and quantification of the 33-mer toxic fraction of gluten, which is harmful to celiac patients), as well as antibodies from human sera obtained from patients diagnosed with celiac disease. The sera were obtained from the Polish Mother’s Memorial Hospital Research Institute (ICZMP). Prior to analysis, the samples underwent in vitro simulated digestion.

**Samples:** For the first part of the experiment, five samples (Table 1) of wheat sourdough bread, differing by recipe, from the Piekarnia Cukiernia Gromulski bakery (Mińsk Mazowiecki, Poland) were used, with yeast bread as the control sample.

In the second part, six different types of wheat sourdough ciabatta rolls differing in terms of the percentage of sourdough and the fermentation time were analyzed (ciabatta I-VI) (Table 2). Three slices from each bread (from both ends and from the middle of the loaf) were cut and air dried at room temperature for 24 h. The slices were then ground together.

**Reagents:** The G12 Gluten Elisa Kit from the Food Allergens Laboratory, Greece was used. The following reagents were purchased from Sigma-Aldrich Co. (St. Louis, MO, USA): angiotensin-converting enzyme (ACE) from rabbit lung (EC 3.4.15.1); N-Hippuryl-His-Leu hydrate (HHL); α-Amylase from porcine pancreas (EC 3.2.1.1) type Vi-B, ≥10 units/mg solid; pepsin from porcine gastric mucosa (EC 3.4.23.1) powder, ≥250 u/mg solid; anti-human IgG-alkaline phosphatase antibody; mouse monoclonal, clone GG-5, purified from a hybridoma cell culture; a ready-to-use solution of alkaline phosphatase yellow (pNPP) liquid substrate system for ELISA. Pancreatin from Kreon Travix 1000, 1000 Ph Eur u lipase, 8000 Ph Eur u amylase and 600 Ph Eur u protease were purchased from Abbott Laboratories GmbH, USA.

**Immunoreactivity analysis by G12 test.** Samples for analysis were prepared according to the protocol for the G12 Gluten Elisa Kit. The only difference was that the supernatant was diluted for the assay. Samples were diluted 1000× for the results to fall within the range of detection of the assay.

**Alpha-Amylase Inhibitory Assay.** This assay was performed using a modified version of the procedure described by Kazeem et al. [14]. The samples were prepared by extracting 0.5 mg of each bread type in 5 mL of distilled water. Starch solution was prepared by dissolving 0.125 g of starch in 25 mL of phosphate buffer. Then, 39 mL of 0.2-M NaH_2_PO_4_ was mixed with 61 mL of 0.2-M Na_2_HPO_4_ and filled up to 200 mL with distilled water. The bread extract was mixed with 4 U/mL α-amylase solution and 0.1-M phosphate buffer (pH 7) and incubated for 10 min at 25 °C with shaking. Next, starch solution was added and the mixture was incubated for 10 min at 25 °C with shaking. The reaction was terminated by adding 1 mL of dinitrosalicylic acid (DNS) reagent. The samples were incubated at 95 °C for 5 min and cooled to room temperature. The reaction mixture was diluted with distilled water (up to 10 mL). The absorbance was measured at 540 nm. The control was a sample of yeast bread. The α-amylase inhibitor activity was calculated as the percentage inhibition using the following equation (Equation (1)):(1)$$\%\;\mathrm{inhibition}=100\;\cdot \frac{{\mathrm{AB}}_{c}-{\mathrm{AB}}_{s}}{{\mathrm{AB}}_{c}}$$
where AB_c_ is the absorbance of the control at 540 nm, and AB_s_ is the absorbance of the sample at 540 nm.

**Angiotensin-Converting Enzyme Activity.** This assay was carried out using the procedure developed by Hernández-Ledesma et al. [5] to establish the best reaction conditions for ACE activity. The following conditions and procedures were followed: 10-mM HHL (dissolved prior to the assay in pH 8.3 buffer (0.2-M phosphate and 0.3-M NaCl)) was mixed with the bread sample (extracted previously with distilled water) and ACE (26 mU/ml of reaction medium, dissolved in glycerol at 50%). This mixture was incubated for 80 min at 37 °C, then 1-N HCl was added to inactivate the enzyme. To extract the reaction product, 1 mL of ethyl acetate was added. The organic layer was taken and dried at 90 °C for 15 min. The dried substance was redissolved in 1 mL of distilled water. Absorbance was determined at 228 nm. The ACE-inhibition percentage was calculated using Equation (2).
(2)$$\%\;\mathrm{IACE}=\frac{\left[\left(A\;-\;B\right)-\left(C\;-\;D\right)\right]}{\left(A\;-\;B\right)}\cdot 100$$
where A is absorbance in the presence of ACE (a sample of yeast bread was used as the control sample), B is the absorbance of the reaction blank (HCl was added before ACE, water was used instead of the sample), C is absorbance in the presence of ACE and the inhibitor and D is the absorbance of the blank sample (water instead of the sample). This equation was used to take into account substances that may have interfered in the assay. If the assay is conducted on samples without possible interference, a simplified equation can be used [5].

**In vitro simulated digestion.** During the second part of the experiment, six samples of ciabatta rolls were analyzed, varying in fermentation time, the proportion of sourdough and the amount of baker’s yeast (Table 2). Yeast bread sourced from the same producer was used as a control.

To assess the effect of the bread on patients suffering from celiac disease, all the samples were first subjected to in vitro simulated digestion. The purpose was to hydrolyze the proteins present in the bread, release the peptides responsible for allergic reactions and mimic the natural process of ingestion.

The experimental conditions were as described by Berti et al. [15]. Each sample was incubated in 0.01-M HCl (pH 2) with pepsin (1:30 *w/w*) for 3 h at 37 °C in a water bath. Next, a solution of 10% NaHCO_3_ was added to neutralize the solution. The reaction mixture was brought up to 10 mL with 1-M Tris HCl and incubated with a pancreatin preparation (1:30 *w/w*) for 2 h at 37 °C in a water bath. The reaction was halted by incubation for 10 min at 90 °C in a water bath. After cooling, the samples were centrifuged at 3500 rpm for 10 min and the supernatant was taken for further analysis.

**Indirect non-competitive ELISA.** This assay was performed using the indirect ELISA method. A microtitre plate (SPL Life Sciences Unibinding) was coated with antigen solution (extracts from simulated digestion diluted 1000 times) in 0.05-M bicarbonate buffer (pH 9.6) and kept overnight at 4 °C. The plate was washed three times in phosphate-buffered saline (PBS) solution with 0.5% Tween 20. The free binding sites were blocked by incubating the plate with a 3% solution of skimmed milk in PBS (pH 7.2) for 2 h. This was followed by removal of the buffer solution, triple rinsing of the plate and further incubation at room temperature for 1 h with the sera of human patients living with celiac disease. The sera, containing antigliadin antibodies, were diluted 100 times with PBS. The plate was washed three times and a diluted solution of anti-IgG antibodies conjugated with alkaline phosphatase was added. After 1 h of incubation, the plate was rinsed using PBS with Tween 20. The bound phosphatase activity was determined in a reaction with pNPP phosphatase substrate using a Multiskan RC (Thermo Labsystems) reader at 405 nm.

**Statistical analysis.** The results were statistically analyzed using the Statistica 13 program, using one-way analysis of variance with Tukey’s post hoc tests to determine the significance of differences at *p* < 0.05. The normality of the distribution was checked using the Shapiro–Wilk test and the assumptions of homogeneity of variance by the Brown–Forsythe test. Statistical studies were performed using Statistica 13.1 (StatSoft, Cracow, Poland). One-way ANOVA analysis was performed with a post hoc Tukey’s test. Differences between means with a 95% (*p* < 0.05) confidence level were considered statistically significant.

### Results Reading

All the plots were created using the RStudio program.

## 3. Results and Discussion

### 3.1. Analysis of Inhibitors of Alpha-amylase

#### 3.1.1. Without in Vitro Digestion. Analysis of Sourdough Bread

The sourdough bread was found to contain more α-amylase inhibitors in comparison to the control (yeast bread). All the analyzed samples showed inhibition of between 11% and 20% (Figure 1A). These results show the retardation of starch degradation. As starch degradation slows, the rate of glucose release falls, resulting in slower increases in the concentration of glucose in the blood after a meal, and as a consequence, moderate secretion of insulin. It can be concluded that the samples had a lower glycemic index (GI). The consumption of products with reduced GI helps maintain healthy body mass. It is also beneficial for people trying to lose weight and crucial for diabetics, for whom minor fluctuations in the concentration of glucose in the blood may cause health problems [16].

Inhibition of enzymes involved in the degradation of starch, mainly α-amylase, but also glucosidases slows the release of glucose and reduces its level in the blood [17].

Slavin et al. proved that the levels of bioactive compounds in bread could be modified using sourdough processes [18]. Flour acidification increases folate content [19], lowers tocopherol and tocotrienol levels [20] and changes thiamin content [21]. It can be concluded that sourdough fermentation can have an influence on the levels of bioactive compounds. However, further research is required to analyze the mechanisms of action of sourdough and its health benefits.

#### 3.1.2. With in Vitro digestion. Analysis of Ciabatta Rolls

Six types of ciabatta rolls were subjected to in vitro digestion and investigated for α-amylase inhibition. This enzyme plays a crucial role in starch digestion. Its activity starts in the oral phase (salivary α-amylase), but its main action is involved in starch hydrolysis in the gastrointestinal phase (pancreatic α-amylase). Freitas at al. [22] have shown that human salivary α-amylase also maintains its activity under in vitro conditions in the gastrointestinal phase, which underlines the importance of this enzyme in digestion and the need for further investigations into its function. Presented results show that the fermentation time of the dough, the level of flour acidification and the presence of yeast have an impact on the inhibition of α-amylase. The highest inhibition level was reached by sample C-VI, with 22 h of fermentation and 50% sourdough. The remaining samples also showed signs of inhibition, although with lower efficiency (Figure 1B). This indicates that higher doses of sourdough, as well as prolonged fermentation, are beneficial. It was also noticed that the addition of bakers’ yeast to the dough increased the ability of sourdough to lower α-amylase activity. It can be concluded that the use of sourdough in bread has the potential to increase the inhibition of α-amylase activity and as a consequence, slow down starch digestion. This suggests that sourdough bread is more suitable for diabetics, like other fermented products for diabetics, e.g., from buckwheat [23].

### 3.2. Analysis of Angiotensin-Converting Enzyme

#### 3.2.1. Without in Vitro Digestion. Analysis of Sourdough Bread

The sourdough bread used in our study had very high levels of IACE when compared to the control (Figure 2A), which was a sample of bread made only with yeast. By inhibiting the angiotensin-converting enzyme, IACE decreases the level of angiotensin II in blood. Angiotensin II indirectly stimulates aldosterone secretion. Reducing the amount of angiotensin II suppresses aldosterone secretion, which in turn blocks the reabsorption of sodium ions and increases their excretion with water. This mechanism lowers blood pressure. Consuming products with high levels of IACE may be beneficial for people suffering from hypertension. The average ACE inhibition of the samples without digestion was 93% and after digestion 59%. There are many literature reports on peptides in foods, including fermented ones, with casein, whey protein, soybean, rapeseed, mushroom, seafood that have ACE inhibiting properties, the effects of which are comparable to drugs. They show digestive resistance in the digestive system [24] and show activity against CaCo-2 cells [25].

The development of food products enhanced with IACE has until recently focused on dairy products [26]. During lactic acid fermentation of cereal goods, high amounts of IACE peptides are released [27]. This is consistent with the results of the present study. Gänzle et al. and Thiele et al. demonstrated that the lack of LAB in straight dough causes low proteolytic activity. The concentration of IACE peptides in bread prepared with a direct dough was 100-fold lower than the concentration in sourdough bread [28,29].

#### 3.2.2. With in Vitro Digestion. Analysis of Ciabatta Rolls

Analyses of ciabatta rolls subjected to in vitro digestion were found to still contain noticeable levels of IACE when compared to the control (Figure 2B). However, when compared to ciabatta samples analyzed without in vitro digestion (Figure 2A), the ACE inhibition level decreased. This suggests that the peptides present in the samples may be partially degraded into smaller peptides, but still maintain their antihypertensive biologic activity.

Similar peptides have been identified in the literature. Studies conducted by Escudero et al. showed that ACE inhibitor peptides from dry-cured ham have good stability against heating [30]. Similar results were obtained for ACE inhibitory peptides derived from tuna cooking juice [31] and from soy protein [32]. Peptides with ACE inhibitory activity were obtained from various fermented products [33]. An antihypertensive effect can be achieved by IACE peptides in vivo if they reach the bloodstream in an active form [34]. After ingestion, the peptides need to resist being completely hydrolyzed by gastrointestinal enzymes and brush border peptidases. They must also be able to go through the intestinal wall while still maintaining their biologic activity. During this process, peptides can be degraded, and their biologic activity may be activated or inactivated. The issue of bioavailability of ACE inhibitory peptides obtained by fermentation in sourdough bread requires further research.

### 3.3. Elisa Immunoreactivity Assay

#### 3.3.1. Without in Vitro Digestion. Analysis of Sourdough Bread

In four out of five examined samples, the content of peptides (33-mer), which are toxic to people with celiac disease increased significantly (Figure 3A). Only in one bread (S-I) did the level of 33-mer decrease.

During lactic acid fermentation with active bacteria in the dough, modifications of flour ingredients connected with the metabolic activity of microbes may occur. Many authors describe the results of research on the effect of lactic fermentation on food ingredients. Particularly interesting is the modification of proteins in terms of their biologic properties. One way to observe whether lactic acid fermentation has taken place is to observe the hydrolysis of gluten proteins, which happens as a result of the action of proteolytic enzymes from microbes participating in the fermentation process. As shown by Thiele et al. [35], proteolysis that occurs during sourdough fermentation is mostly related to the activation of enzymes present in cereal mediated by the pH. A drop in pH, caused by the activity of LAB, activates endogenous flour enzymes which also initiate gluten protein hydrolysis. Those processes cause the release of peptides, which can be harmful to celiac patients, due to the presence of toxic epitopes.

The increase in the immunoreactivity of the five analyzed samples compared to the control suggests that there was a high degree of structural modification in the flour proteins as a result of dough fermentation. Leszczyńska et al. [36] found that the immunoreactivity of bread made from fermented flour was lower than that of bread made from non-fermented flour. However, different antibodies (human anti-gliadin antibodies, serum S 875; rabbit anti-QQQPP antibodies) were used in that study, and the differences in their results may be caused by the reactions of individual patients to the gluten proteins in bread. Previous studies by De Angelis et al., 2006 [37], Di Cagno et al., 2002 [38], Leszczynska et al., 2009 [39], Rizzello et al., 2006 [40] and Rollan et al., 2005 [41] suggested that lactic acid bacteria (LAB) was an effective tool for the reduction of wheat allergens. Further experiments were therefore conducted with samples subjected in vitro digestion (Figure 3B).

#### 3.3.2. With in Vitro Digestion. Analysis of Ciabatta Rolls

Based on the results (Figure 3B), it can be seen that the addition of sourdough to the analyzed ciabatta rolls lowered the immunoreactivity of wheat flour proteins. Acidification and proteolysis of flour decreased the level of G12, a peptide harmful to patients suffering from celiac disease, by half. On average, only 54% of this peptide remained active. This action can be explained by the proteolytic activity of the lactic acid bacteria present in the sourdough microflora. Given sufficiently long fermentation time for the development of the microflora, even 9% sourdough produced satisfactory results, with a drop in immunoreactivity comparable to most of the analyzed samples. Similar results of studies on the reduction of immunoreactivity of peptides harmful to celiac disease patients, based on acidification and proteolysis of flour samples, have been received by other researchers [42].

The data suggest that sourdough bread may be more susceptible to enzymes in the digestive system. This could result in better tolerance by immunosensitive individuals. However, this is only a hypothesis which must be confirmed by research, including in vitro simulated digestion and in vivo studies. Based on the results of the present study, the examined samples of bread cannot be incorporated into the diets of celiac patients, as they still present considerable immunoreactivity.

Di Cagno et al. [43] studied the ability of lactobacilli in sourdough to hydrolyze wheat prolamins and proposed a new formula for sourdough bread suitable for celiac patients. In a comparison of various types of bread, celiac patient showed better tolerance to the sourdough bread than straight dough bread. On this basis, the application of selected strains of LAB and long fermentation times can be regarded as useful and safe ways of lowering the level of human intolerance to wheat proteins.

### 3.4. Immunoreactivity Analysis With Human Antibodies

A more detailed study was conducted with human antibodies. Although the patient sera showed diverse reactions depending on the individual’s susceptibility to gluten, decreased antigenicity of wheat flour proteins was observed in all cases (Figure 4). The average remaining immunoreactivity was 58% for serum 813% and 68% for serum 1493. In the case of serum 222, antigenicity dropped markedly, to about 12% on average. This may indicate that the patient is able to tolerate wheat sourdough products.

Products made from wheat flour are consumed after thermal processing. Baking is considered to increase allergenicity due to the creation of new epitopes in Maillard reactions [44,45]. This may explain in part the higher immunoreactivity of the tested samples. Particular modifications to the baking parameters are closely related to the individual responses of patients. In the case of the least sensitive patient, it is clearly visible that increasing the proportion of sourdough improved their tolerance to wheat bakery products. A comparison of the samples similarly suggests that the addition of baker’s yeast is beneficial even with shorter fermentation times. Although LAB show the ability to diminish the immunoreactivity of gluten, it should be remembered that each bacterial strain acts in a different manner and that its action can differ significantly when co-habiting with other strains [39]. Despite the fact that the immunoreactivity of ciabatta rolls with acidified flour dropped compared to the control without sourdough in the formula, more research is required to find the most suitable recipe and production process, as well as to prove that the remaining immunoreactivity in the bread represents no/Low risk for immunosensitive individuals.

## 4. Conclusions

This study on immunoreactivity confirmed the initial hypothesis that the presence of sourdough in wheat bread alters the structure of the flour’s peptides over the course of fermentation. Along with fermentation of lactic acid, changes in the components of the flour occur due to the metabolic activity of microorganisms. Both the hydrolysis of gluten peptides by proteolytic enzymes and the activation of endogenous enzymes present in the flour were caused by a drop in pH. Thanks to these processes, the samples were susceptible to toxicity-reducing digestion enzymes. Based on the results presented here, it can be concluded that baked wheat goods produced with sourdough have the potential to decrease immunoreactivity and can be tolerated well by patients with moderate gluten intolerance. It should be remembered that immune responses vary depending on the patient. The types of wheat bread studied may be recommended for patients in the high-risk group and families with a history of celiac disease. It can also be considered for patients with hypertension and, to a limited extent, diabetes. Based on hereby research it can be concluded that the use of sourdough in bread has the potential to increase the inhibition of α-amylase activity (in the range 10%–90%) and as a consequence, slow down starch digestion. This suggests that sourdough bread is more suitable for diabetics.

Consuming products with high levels of IACE may be beneficial for people suffering from hypertension. The average ACE inhibition of the samples without digestion was 93% and after digestion 59%.

## Figures and Tables

**Figure 1 foods-09-00656-f001:**
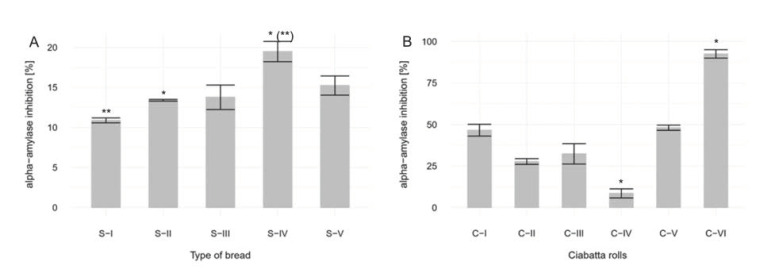
(**A**). Percentage of α-amylase inhibition in different types of sourdough bread compared to yeast bread, in which α-amylase inhibition is 0%. Samples marked with * and ** are statistical significance at the *p* < 0.05 level. (**B**). Percentage of α-amylase inhibition in ciabatta rolls compared to yeast bread, where α-amylase inhibition is 0%. Samples marked with * are statistical significance at the *p* < 0.05 level.

**Figure 2 foods-09-00656-f002:**
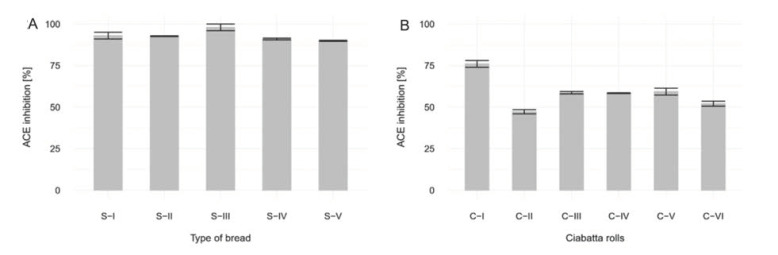
(**A**). Percentage of angiotensin-converting enzyme (ACE) inhibition in different types of sourdough bread compared to yeast bread, for which inhibition is equal to 0%. Results of ANOVA test *p* > 0.05. (**B**). Percentage of ACE inhibition in different types of ciabatta rolls after digestion compared to yeast bread, for which inhibition is equal to 0%. Results of ANOVA test *p* > 0.05.

**Figure 3 foods-09-00656-f003:**
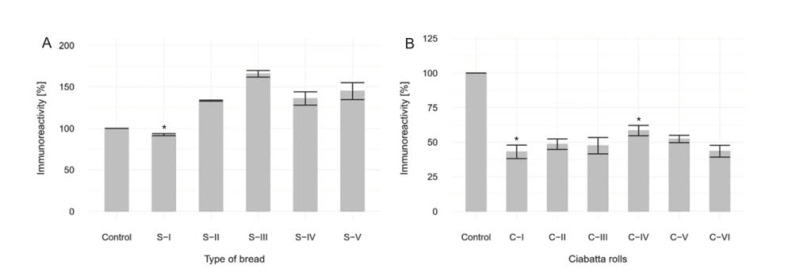
(**A**). Change in immunoreactivity of wheat flour proteins for different types of sourdough bread relative to yeast bread. Samples marked with * are statistical significance at the *p* < 0.05 level. (**B**). Change in immunoreactivity of wheat flour proteins for different types of ciabatta rolls relative to yeast bread. Samples marked with * are statistical significance at the *p* < 0.05 level.

**Figure 4 foods-09-00656-f004:**
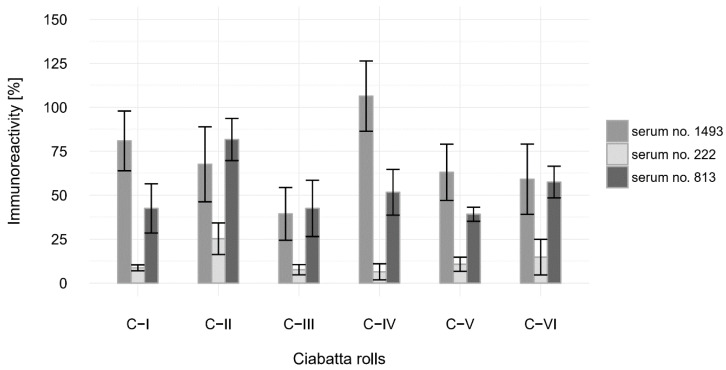
Immunoreactivity changes in wheat proteins for different types of ciabatta rolls relative to yeast bread (100%) in a reaction with antibodies present in the human sera of three immunosensitive patients.

**Table 1 foods-09-00656-t001:** Analyzed samples of wheat sourdough bread.

S-I	Wheat sourdough bread	Wheat flour (0.75% ash content), wheat sourdough, water, barley malt, baker’s yeast, salt
S-II	Wheat-rye sourdough bread with flax and sunflower seeds	Wheat flour (0.75% ash content), rye flour (0.72% ash content), wheat sourdough, water, baker’s yeast, flax and sunflower seeds, plant oil, salt
S-III	Wheat-rye sourdough bread fermented spontaneously with wholegrain rye flour	Wheat flour (0.75% ash content), rye flour (2.00% ash content), wheat sourdough, water, barley malt, baker’s yeast, salt
S-IV	Ciabatta type 1	Wheat flour (0.55% ash content), wheat sourdough, water, baker’s yeast, salt
S-V	Ciabatta type 2	Wheat flour (0.55% ash content), wheat sourdough, water, baker’s yeast, salt
Control	[prolonged fermentation]	Wheat flour (0.55% ash content), water, baker’s yeast, salt

Samples composition is given in Table 1.

**Table 2 foods-09-00656-t002:** Analyzed samples of ciabatta rolls.

Sample	Fermentation Time (h)	Sourdough (%)	Yeast (%)
C-I	22	9	1
C-II	22	15	0
C-III	3	15	4
C-IV	22	30	0
C-V	3	30	4
C-VI	22	50	0

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
