# Peer review of "The Inhibition of Amylase and ACE Enzyme and the Reduction of Immunoreactivity of Sourdough Bread"

_foods, 2020, doi:10.3390/foods9050656_

Round 1

Reviewer 1 Report

The manuscript demonstrates the possible health-promoting effect of sourdough bread. The topic is interesting, however, a more in-depth comprehensive analysis of the results must be done. Limitations for the present study should be discussed.

Title: The title of the article is very ambitious and the authors might consider modifying it.

Lines 32-33: Check if the style of the reference is correct.

Lines 101-104: The authors should add detailed recipes (with the composition) of bread

Line 106: What were the conditions of drying of bread?

Line 108: Add country the Food Allergens Laboratory.

Line 115: Add country for ABBOTT PRODUCTS

Line 188: Add details for the dilution.

Line 123: Add details for the phosphate buffer.

Line 124: Were the mixtures incubation on a shaker?

Lines 151-153: : The authors should add the detailed recipies (with the composition) of ciabatta rolls

Line 181: The explanation of statistical analysis of the data is missing

All figures: I would recommend adding letters that show the statistical difference between the samples + Add what the abbreviations of the samples mean.

Line 194-196: Add a reference

Lines 192-205: The discussion is insufficient.

Lines 213-227: The discussion is insufficient.

Lines 219-220: “…the fermentation time of the dough, the level of flour acidification and the presence of yeast…” – support the sentence with the relevant data.

Lines 236-237: Where is the control?

Lines 236-242: The discussion is insufficient.

Line 257-261: The discussion is insufficient.

Lines 279-281: The discussion is insufficient.

Line 313:  Add comparison to literature

Conclusion: should be improved

Author Response

Thank you very much for the comments regarding the review. All comments have been taken into account when correcting the text.

Title: The title of the article is very ambitious and the authors might consider modifying it.

The title is changed “The inhibition of amylase and ACE enzyme,  and the reduction of immunoreactivity of sourdough bread”

Lines 32-33: Check if the style of the reference is correct.

The style of references id corrected.

Lines 101-104: The authors should add detailed recipes (with the composition) of bread

Detailed recipes and composition of bread are added in the table 1.

Line 106: What were the conditions of drying of bread?

It is added in the text.

Line 108: Add country the Food Allergens Laboratory.

It is added.

Line 115: Add country for ABBOTT PRODUCTS

It is added.

Line 118: Add details for the dilution.

It is added in the text.

Line 123: Add details for the phosphate buffer.

The details for the phosphate buffer are added.

Line 124: Were the mixtures incubation on a shaker?

It is e explained and added.

Lines 151-153: : The authors should add the detailed recipies (with the composition) of ciabatta rolls

The recipies are added in the table 2, but the details are the manufacturer secret.

Line 181: The explanation of statistical analysis of the data is missing

The statistical analysis is added and the figures are changed.

Line 194-196: Add a reference

The reference is added.

Lines 192-205: The discussion is insufficient.

The discussion is improved.

Lines 213-227: The discussion is insufficient.

The discussion is improved.

Lines 219-220: “…the fermentation time of the dough, the level of flour acidification and the presence of yeast…” – support the sentence with the relevant data.

This data are in the table 2.

Lines 236-237: Where is the control?

The control sample is the bread with yeast without sourdough.

Lines 236-242: The discussion is insufficient.

The discussion is improved.

Line 257-261: The discussion is insufficient.

The discussion is improved.

Lines 279-281: The discussion is insufficient.

The discussion is improved.

Line 313:  Add comparison to literature

It is added.

Conclusion: should be improved

The conclusions are improved.

Reviewer 2 Report

After making major corrections, I think it will be particularly interesting for scientists working on reducing the spread of diabetes and treating celiac disease.

I believe that the introduction is too extensive (as a research type of article). It should be reduced by approx. 30% of the text.

 „All the results were analyzed using the RStudio program.” - how it was analyzed?

Figure 1 – What are S-I, S-II…? Samples? Why were the abbreviations not described in the methodology and the breads bought were generally characterized? I miss the detailed analysis of the samples tested - above all the recipes and the method of production.

Figures 1 – 6: Optically the drawings are unsightly, simple and too big for the values they present. They can be combined or presented in a table. Why was ANOVA not carried out?

What's new and innovative in these studies?

I don't know if this is a matter of formatting, but in this form the manuscript title is illegible. Please edit and shorten it

Author Response

Thank you very much for the comments regarding the review. All comments have been taken into account when correcting the text.

I believe that the introduction is too extensive (as a research type of article). It should be reduced by approx. 30% of the text.

The introduction is reduced.

 „All the results were analyzed using the RStudio program.” - how it was analyzed?

…the results were created …It is changed in the text.

Figure 1 – What are S-I, S-II…? Samples? Why were the abbreviations not described in the methodology and the breads bought were generally characterized? I miss the detailed analysis of the samples tested - above all the recipes and the method of production.

The details connected with the samples, their recipes and the method of production are in the table 1 and table 2.

Figures 1 – 6: Optically the drawings are unsightly, simple and too big for the values they present. They can be combined or presented in a table. Why was ANOVA not carried out?

The drawings are improved and the statistical analysis is added.

What's new and innovative in these studies?

The innovation is demonstrating the healthy operation of commercial bread sourdough. Such bread may be beneficial for people with diabetes, hypertension and hypersensitivity to gluten.

I don't know if this is a matter of formatting, but in this form the manuscript title is illegible. Please edit and shorten it

The title is changed.

Reviewer 3 Report

The topic of foods-797548 is interesting and fits well the scope of Foods. However, extensive amendments are required before it can be accepted.

(1) The authors have to polish their English writing extensively.

(2) Statistical analysis is missing.

(3) As human sera were used, ethical clearance should be provided.

(4) “Angiotensin-Converting Enzyme Activity.” This part of study is quite meaningless as so far the absorption of the substances in the bread remains unknown.

Author Response

Thank you very much for the comments regarding the review. All comments have been taken into account when correcting the text.

  • The authors have to polish their English writing extensively.

The English is checked by native speakers - specjalist in biotechnology and food sciences.

(2) Statistical analysis is missing.

The statistical analysis is added.

(3) As human sera were used, ethical clearance should be provided.

All research wea conducted on the base of agreement of the Ethical Comitee by the Polish Mother’s Memorial Hospital Research Institute (ICZMP).

(4) “Angiotensin-Converting Enzyme Activity.” This part of study is quite meaningless as so far the absorption of the substances in the bread remains unknown.

This part of study requires further investigation – it is explained in the text.

Reviewer 4 Report

I understand that you used a local bakery who prepared and fermented your samples, however I would like to know the different combinations. Plus how long you allowed the samples to ferment. I feel that, that information is important. 

In regards to your statistics. What type of statistics did you run? You need to indicate that in the methods section. 

You show error bars on your graphs what do they represent? SEM, STD? There is no explanation of what they are. 

Author Response

Thank you very much for the comments regarding the review. All comments have been taken into account when correcting the text.

I understand that you used a local bakery who prepared and fermented your samples, however I would like to know the different combinations. Plus how long you allowed the samples to ferment. I feel that, that information is important. 

A: Available information is given in tables 1 and 2

In regards to your statistics. What type of statistics did you run? You need to indicate that in the methods section. 

A: The details of statistical anaylis are in the text.

Statistical studies were performed using Statistica 13.1 (StatSoft, Cracow, Poland). One-way ANOVA analysis was performed with a post-hoc Tukey’s test. Differences between means with a 95% (P<0.05) confidence level were considered statistically significant.

You show error bars on your graphs what do they represent? SEM, STD? There is no explanation of what they are. 

A: This data are added in the text.

Round 2

Reviewer 2 Report

The article was corrected according to the reviewer's comments.

Author Response

Thank you very much. I am sending the corrected paper according th the comments.

Best regards,

Joanna Leszczynska
